# The role of obesity in physiological stress, balance, and proprioception during repetitive manual material handling tasks

Sergio A. Lemus[1], Jaron Mohammed[1], Cheng-Bang Chen[2], Thomas M. Best[3,4], Eduard Tiozzo[5], Francesco Travascio[1,3,6]*

**1** Department of Mechanical and Aerospace Engineering, University of Miami, Coral Gables, Florida, United States of America, **2** Department of Industrial and Systems Engineering, University of Miami, Coral Gables, Florida, United States of America, **3** Department of Orthopaedics, University of Miami Miller School of Medicine, Miami, Florida, United States of America, **4** Sports Medicine Institute, University of Miami Miller School of Medicine, Miami, Florida, United States of America, **5** Department of Physical Medicine and Rehabilitation, University of Miami Miller School of Medicine, Miami, Florida, United States of America, **6** Max Biedermann Institute for Biomechanics at Mount Sinai Medical Center, Miami Beach, Florida, United States of America

* f.travascio@miami.edu (FT)

## Abstract

Manual laborers often experience fatigue-related incidents, which increase their risk of balance disturbances and falls. Previous research indicates that obese individuals may reach critical fatigue levels during repetitive lifting. This study examines whether this BMI-based fatigue pattern also applies to other manual handling activities. Therefore, assessing balance impairment under high physiological stress conditions will help quantify the increased fall risk in obese individuals. Thirty participants performed carrying and pushing/pulling tasks, with weights determined using the Liberty Mutual Equations to align with NIOSH criteria. Balance tests were conducted before and after each task. A two-way ANOVA compared energy expenditure rate (EER) across BMI classifications and sex, while a mixed-effects model analyzed the effects of EER, BMI, and sex on balance and proprioception tests. Results indicated a positive correlation between BMI and EER for both carrying (p = 0.003) and pushing/pulling (p = 0.013). In the mixed-effects model, BMI (p = 0.032) and EER (p = 0.037) were positively correlated with knee proprioception loss, whereas EER was negatively correlated with balance (p = 0.020). These findings confirm that obese individuals face critical fatigue levels, as well as impaired proprioception and balance, during repetitive handling tasks.

## Introduction

Nearly 4 out of 10 adults over the age of 20 are obese in the United States (U.S.) [1]. The healthcare costs attributed to this epidemic amount to as much as $210 billion

**Data availability statement:** All relevant data are within the manuscript and its Supporting Information files.

**Funding:** This work was supported by the University of Miami under the Laboratory for Integrative Knowledge (U-LINK) Predoctoral Fellowship in Interdisciplinary Research (Award Number U-LINK 23-4095) and the 2024 Summer Academic Enhancement Research Fellowship The funders had no role in study design, data collection and analysis, decision to publish, or preparation of the manuscript.

**Competing interests:** The authors have declared that no competing interests exist.

per year. Additionally, obesity negatively impacts workplace productivity, estimated at $12 billion annually [2]. Overall, obese and overweight workers make up over 60% of the U.S. workforce [1] and are 68% more likely to sustain workplace injuries compared to workers of normal weight [3]. In 2013, the Transport, Warehouse, and Utilities (TWU) Super Sector had the highest prevalence of obesity at 34.2% [4,5] and the highest rate of musculoskeletal injury [6,7] across all occupational groups. The organizational structure of work in the TWU sector involves long and irregular work hours, time pressures, and nonstandard work arrangements[8]. These factors can lead to work-related incidents due to distraction and fatigue [8,9]. Fatigue, in particular, is associated with disturbances in cognition [8,9] and balance [10–12], both of which considerably increase the risk of falls [13,14].

One-fifth of all occupational claims are slips, trips, or falls [15]. Obese individuals have a lower percentage of fatigue-resistant skeletal muscle fibers [16], making them more likely to lose balance and fall [17]. Balance is influenced by numerous factors, including proprioception [14,18], which is the sense of body position, force and movement [19]. Several studies indicate that assessing knee proprioception can help quantify the risk of falls due to balance impairment [14,20]. Therefore, assessing loss of balance and proprioception can be beneficial for measuring the increased risk of falling in obese individuals.

Despite the link between obesity, fatigue, and injuries due to the risk of falling in occupational workers, work tasks are still being designed using anthropometric characteristics from normal body mass index (BMI) individuals [21,22]. The National Institute of Safety and Health (NIOSH), the federal agency responsible for making recommendations for the prevention of work-related injuries in the U.S., established the Revised NIOSH Lifting Equation (RNLE) in 1993 [23]. The RNLE calculates recommended weight limits (RWL) to keep compression loads on the lower back (biomechanical criterion) and the energy expenditure (physiological criterion) within safety limits during manual lifting. Experimental coefficients accounting for work geometry, frequency, and duration of the task are included in the RNLE. Notably, the calculated RWL is designed to maintain a person's energy expenditure rate (EER) within a safety limit of 4.7 kcal/min [23]. However, this calculation does not consider physical characteristics of the participant such as clinically elevated BMI. In fact, we have shown that obese individuals lifting at RWL surpass the physiological safety limit established by the RNLE, compared to overweight and normal-weight individuals [24].

Similarly to the RNLE, the maximum acceptable loads (MAL) for other manual material handling (MMH) activities such as pushing, pulling, and carrying were first estimated by the Liberty Mutual MMH Tables [25]. These standards were developed based on the RNLE psychophysical criterion [26], aiming to ensure that more than 75% of the female population could perform a specific task without overexertion [27]. The Liberty Mutual MMH Tables were recently replaced by fourteen equations that summarize the work at the Liberty Mutual Research Institute [28]. Similar to the RWL calculation, the Liberty Mutual MMH Equations calculate MAL given geometrical and frequency parameters for the given push, pull or carry task. However, they do not consider the worker's anthropometric characteristics.

Characterizing the effects of clinically elevated BMI on physiological stress parameters during manual activities beyond lifting is essential to determine whether this pattern of dangerous fatigue levels in obese individuals persists. In addition, evaluating the loss of proprioception and balance when safety thresholds for physiological stress are surpassed could aid with assessing the increased risk of fall for obese individuals. The goal of this study is to evaluate the role of BMI on physiological stress during carrying, pushing, and pulling, and quantify how such levels of physiological stress may impact proprioception and balance. It was hypothesized that for the MMH tasks tested, obese individuals would surpass the NIOSH threshold and when safety limits are exceeded, due to fatigue, obese participants would experience a significant loss of balance and proprioception,

## Materials and methods

### Subject recruitment

The methods used in this study were approved by the University of Miami Human Subject Research Office with Internal Review Board number #20211175 by written consent, and all experiments were performed in accordance with relevant guidelines and regulations. Recruitment started on 19/03/2024 and ended on 11/06/2024. All participants were informed of the study procedures, provided written consent, and were given the option to withdraw at any time. Participants were recruited from the local county through advertisement flyers posted in common areas surrounding the main campus. The inclusion criteria consisted of any adult with no current or prior history of musculoskeletal injury or any medical condition, such as chronic cardiopulmonary conditions or peripheral neuropathy, that would prevent them from participating in the physical activities required for this experiment. A total of 30 participants were enrolled, stratified into 3 groups of 10 (5 males and 5 females) across three BMI classifications [29,30]: normal (18.5 to < 25 kg/m$^2$), overweight (25 to < 30 kg/m$^2$), and obese (>30 kg/m$^2$). The ethnic and racial composition of the participants followed as closely as possible the demographics of the local county [31].

### Work geometry

Participants completed two MMH tasks: a carrying task, and a combined pushing and pulling task. The MALs for the specified distances and frequencies of each task were determined using the Liberty Mutual MMH Equations [28,32]. This experimental routine was conducted in a single session (total time 120 mins), with 30-minute rest intervals between the carrying and pushing-pulling tasks. The order of the tasks was randomized. During carrying task, participants carried a 25.7 lbs. box (20"x10"x10") by grabbing handles positioned 36" from the floor and walking 25 ft. in a straight line. This carrying action was repeated every 15 seconds for a total duration of 15 minutes. During the pushing-pulling task, participants pushed a weighted sled, with their hands placed 35" from the floor, along a straight 25-foot path. After 15 seconds from the initial push, participants pulled the sled back to its original position along the same path. This push-pull action was also repeated for 15 minutes. The force required to push and pull the sled was 19 lbs., measured using a scale balance on the smooth surface employed for testing, following the procedure of Matthew et al. [33]. Table 1 provides the specific values used in the MAL calculations.

### Metabolic measurements

Metabolic data during the MMH tasks were acquired via a CardioCoachCO2 metabolic cart (KORR, Utah, U.S.) in conjunction with a heart rate monitor (Polar, Kempele, Finland). The cart paralleled participants' movements to ensure calorimetry readings were recorded for the entire duration of each MMH task. After a standardized automatic calibration, the equipment measured volume of carbon dioxide production ($VCO_2$), volume of oxygen consumption ($VO_2$), and (heart rate) HR. Values for $VO_2$ and $VCO_2$ were used to determine the EER according to Weir JB [34], as shown

**Table 1. MAL values for MMH tasks.** MAL for both tasks in this study were established using the Liberty Mutual MMH Equations. MAL values which included at least 75% of the female population were used to limit overexertion and maintaining the task within the outlined safety limits.

| Variable | Carrying | Pushing | Pulling |
|---|---|---|---|
| *Frequency (actions per minute)* | 4 | 4 | 4 |
| *Initial Force (lbs.)* | N/A | 30 | 30 |
| *Horizontal Distances (ft)* | 25 | 25 | 25 |
| *Vertical Hand Height (in)* | 36 | 35 | 35 |
| *Sustained Force (lbs.)* | 25.7 | 19.0 | 19.0 |
| *Percentage of Female Population (%)* | 79 | 78 | 77 |

in equation (1). Specifically, time-dependent measurements for EER and HR were reported with sampling at 15 second intervals for each participant.

$$EER \left[\frac{kcal}{min}\right] = 3.94 \cdot VO_2 \left[\frac{ml}{min}\right] + 1.11 \cdot VCO_2 \left[\frac{ml}{min}\right]$$

(1)

### Proprioception and balance testing

Four tests were conducted to assess participants' proprioception and balance during the MMH tasks in this study: the knee proprioception test [35,36], the Sharpened Romberg test [37], the Functional Reach test [38,39], and the Sit-to-Stand test [40,41].

To measure knee proprioception, a device using two 9-degree-of-freedom ICM-20948 [42] Adafruit Inertial Measurement Units (IMU) (Adafruit, New York, U.S.) was built. The IMUs were attached to a 3D-printed mechanism designed in Fusion 360 (v16.8, Autodesk, San Francisco, U.S.) that locked the knee brace at 40º and 70º of knee flexion. The angle of rotation was measured with a gait monitoring system previously validated for the ICM-20948 IMUs [43–46]. Participants, blindfolded and guided on a smooth and pattern-less surface, were first taken to a knee flexion angle of 40º, followed by 70º. They were then instructed to reproduce the demonstrated 40º angle from the 70º flexion while wearing noise-cancelling headphones.

After completing the knee proprioception test, participants proceeded to the Sharpened Romberg test [37], in which they stood in a tandem position with arms crossed and eyes closed. They were barefoot to limit any advantages or disadvantages from footwear [47,48]. Participants were instructed to maintain balance for as long as possible, up to 60 seconds per trial. The timer was stopped if a participant lost the tandem position, uncrossed their arms, or opened their eyes during the test. For the Functional Reach test, participants were barefoot and instructed to reach as far forward as possible in a uniaxial horizontal plane without tipping over [38,39]. The horizontal distance between participants' initial upright position and the farthest point reached was recorded. In the Sit-to-Stand test, participants performed as many sit-to-stand motions as possible within 30 seconds [40,41]. They were instructed to sit with their full weight on the chair, keeping their arms crossed and feet parallel for the duration of the test. The sequence of the experimental procedure, including the geometrical setup for both MMH tasks performed in the study (carrying and pushing/pulling), as well as images of each test measuring balance and proprioception before and after MMH tasks were completed, is shown in Fig 1.

### Statistical analysis

All statistical analyses were conducted using Minitab (21.1.1, Minitab LLC, Pennsylvania, U.S.) and R (4.3.2, R Foundation for Statistical Computing, Vienna, Austria). A seven-point moving average was calculated for EER and HR at each time interval within each BMI classification using Python® (3.11, Python, Delaware U.S.). A 5-minute stabilization period was considered when investigating the statistical differences across metabolic parameters [24,49]. The level of significance for all

 

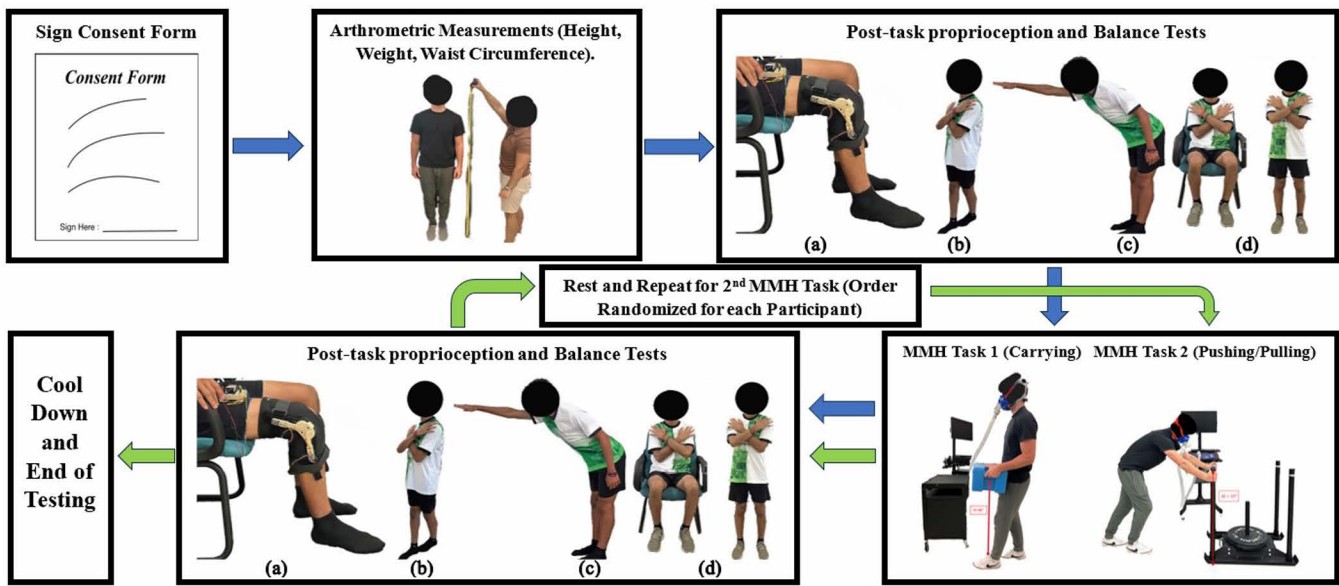

**Fig 1. Experimental procedure pipeline.** Chronological order of activities instructed to participants upon arrival at the testing site. Images of each proprioception and balance test performed are shown: (A) the knee proprioception test with the participant at a flexion angle of 70º, (B) the Sharpened Romberg test with the participant standing in the tandem position, (C) the Functional Reach test where the participant reaches forward, and (D) the Sit-to-Stand test with the participant alternating between sitting and standing.

statistical tests was set at 95% (α = 0.05). Descriptive statistics were used to report means and standard deviations for each metabolic variable (EER and HR) as well as the balance and proprioception test values (knee proprioception, Sharpened Romberg, Functional Reach, and Sit-to-Stand). Grubb's test was performed to detect outliers test within the dataset. A two-way analysis of variance (ANOVA), followed by a Tukey's post-hoc test, was conducted to compare the steady-state values of EER and HR. The factors were sex (15 males and 15 females) and BMI classification (10 normal, 10 overweight, and 10 obese). The assumptions of ANOVA were checked by assessing the homogeneity of variances using Levene's test and the normality of residuals using the Shapiro-Wilk test. Additionally, linear regression models were used to investigate the correlation of BMI with the steady-state values of EER, HR, and waist circumference. Furthermore, to account for fixed and random effects, linear mixed-effects models [50,51], using the R package lme4 [52,53], were employed to assess the influence of BMI, EER, and sex on each balance and proprioception test. Participants and MMH tasks were treated as random effects. Coefficients, standard errors for fixed effects, and intercept interaction were reported, and formulas were developed for significant effects. Lastly, a post-hoc power analysis was conducted using G*Power (version 3.1) [54] to assess the statistical power based on the sample size for each metabolic variable, considering sex and BMI as the main effects, and for balance/proprioception values, considering EER and BMI as the main effects.

## Results

### Participants' demographics and occupation

The sample (n = 30) comprised of 50% males (n = 15), and 50% females (n = 15), mean age of 26.5 years (SD = 8.9), BMI of 29.0 kg/m² (SD = 9.0), and waist circumference of 93.2 cm (SD = 18.2 cm), see Table 2. The relationship between BMI and waist circumference showed a strong positive correlation (r² = 87.5%, p < 0.001), confirming that participants with elevated BMI had increased body fat percentage rather than muscle mass [55]. The overall racial/ethnic distribution was as follows: 46.7% Hispanic/Latino (n = 14), 26.7% white, not Hispanic/Latino (n = 8), 13.3% Asian (n = 4), 10.0% Black (n = 3),

**Table 2. Age and anthropometric data of participants. All the data are reported as mean ± standard deviation.**

| Variable | Sex | Age (years old) | BMI (kg/m²) | Waist (cm) |
|---|---|---|---|---|
| Normal BMI (n = 10) | Male (n = 5) | 20.6 ± 6.5 | 22.7 ± 1.5 | 83.1 ± 5.5 |
| | Female (n = 5) | 24.6 ± 9.2 | 19.3 ± 2.5 | 72.4 ± 6.8 |
| | *Subtotal* | *22.6 ± 6.5* | *21 ± 2.7* | *77.7 ± 8.1* |
| Overweight BMI (n = 10) | Male (n = 5) | 25.8 ± 5.7 | 27.3 ± 1.7 | 93.2 ± 6.9 |
| | Female (n = 5) | 28.8 ± 15.3 | 27.6 ± 1.7 | 93.6 ± 4.6 |
| | *Subtotal* | *27.3 ± 11.0* | *27.5 ± 1.6* | *93.4 ± 5.5* |
| Obese BMI (n = 10) | Male (n = 5) | 28.4 ± 5.6 | 44.6 ± 9.0 | 123.6 ± 19.3 |
| | Female (n = 5) | 30.8 ± 10.5 | 35.5 ± 1.6 | 93.4 ± 7.3 |
| | *Subtotal* | *29.6 ± 8.0* | *38.6 ± 8.8* | *108.5 ± 21.0* |

and 3.3% two or more races/ethnicities (n = 1). The overall labor force distribution of the subjects was as follows: 70.0% education, 13.3% healthcare, and 16.7% service.

## Effects of BMI and sex on metabolic parameters during MMH tasks

Fig 2 shows the values of the metabolic parameters, including EER and HR, over the entire 15-minute period for each MMH task. The average EER varied by BMI classification for both carrying and pushing/pulling tasks, with obese individuals experiencing higher values compared to those of normal weight. Significant BMI differences in HR were observed only during the carrying task. Notably, after the first 5 minutes, the values of EER for obese participants (5.34 kcal/min) surpassed the NIOSH safety threshold (4.7 kcal/min) by 13.6% during the carrying task, while the EER for overweight (4.51 kcal/min) and normal-weight participants (3.25 kcal/min) remained below the threshold. In contrast, for the pushing/pulling task, all BMI classifications exceeded the NIOSH safety threshold. Obese (7.67 kcal/min) and overweight participants (6.51 kcal/min) exceeded the threshold by 63.2% and 38.5%, respectively, compared to normal-weight participants (5.53 kcal/min), who surpassed it by 17.7%.

Table 3 presents the results of the two-way ANOVA analysis comparing the average steady-state values of EER and HR across sex and BMI classifications for each MMH task. During the carrying task, there was no significant interaction between BMI and sex. However, EER was significantly influenced by BMI (p = 0.003) and sex (p = 0.013) independently. Specifically, the steady-state EER for obese participants was significantly higher than for normal-weight participants by 64.3%, and males had a significantly higher EER than females by 31.6%. Similarly, for the pushing/pulling task, there was no significant interaction between BMI and sex. However, significant interactions were found for EER with BMI (p = 0.013) and EER with sex (p = 0.039). The EER for obese participants was significantly higher than for normal-weight participants by 38.7%, and males had a significantly higher EER than females by 19.9%. Regarding HR, during the carrying task, BMI had a significant effect (p = 0.038), while sex did not. No significant effects on HR were observed during the pushing/pulling task. A post-hoc power analysis indicated power ≥ 95% when inferring statistical differences in EER and HR across BMI groups and sex.

Additionally, simple linear regression models were used to explore the relationships between BMI and EER, and HR. For the carrying task, significant correlations were found between BMI and EER (r² = 68.9%, p < 0.001) and between BMI and HR (r² = 20.0%, p = 0.013). In contrast, for the pushing/pulling task, a significant correlation was found only between BMI and EER (r² = 37.6%, p < 0.001), with no significant correlation between BMI and HR.

## Effects of EER, BMI, and sex on balance and proprioception

Descriptive statistics for each balance and proprioception test, categorized by MMH task and BMI classification, are provided in Table 4. Values represent the mean differences between test measurements taken before and after the MMH tasks. Results of the linear mixed-effects model are shown in Table 5. No significant interactions were observed for the

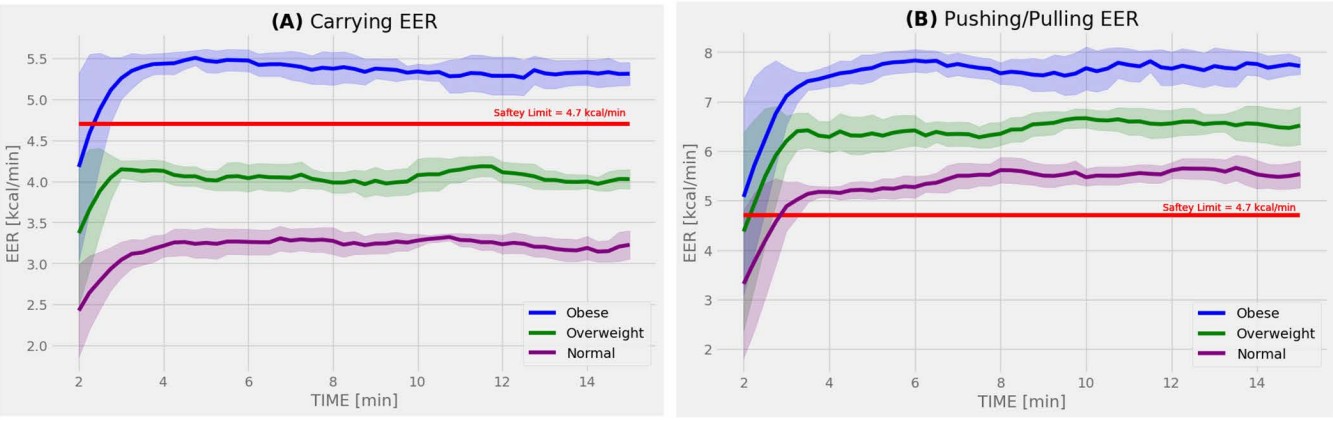

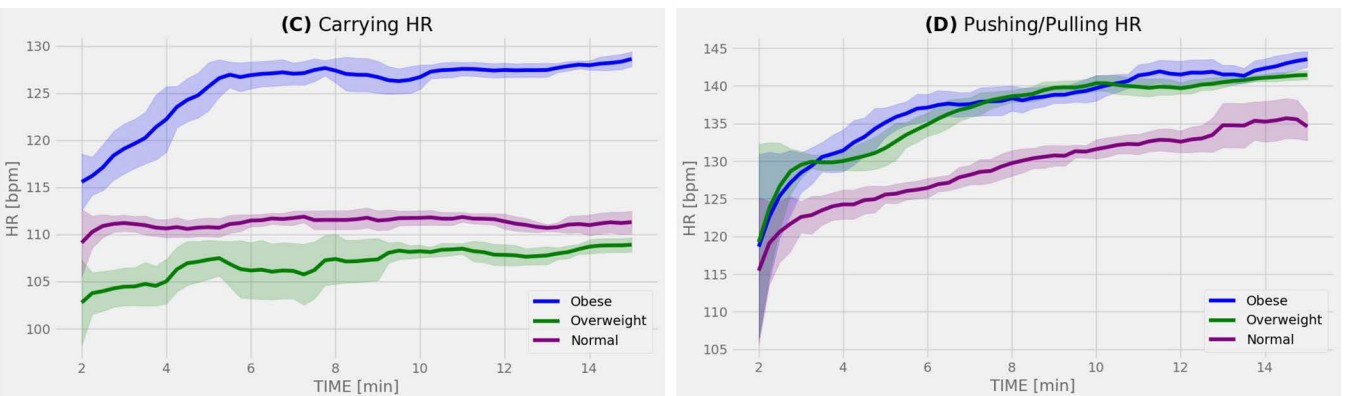

**Fig 2. Time-dependent metabolic parameters across testing period for each MMH task.** Subjects were grouped into their corresponding BMI classification: blue for obese, green for overweight, and purple for normal. The solid red line red represents the 4.7 kcal/min NIOSH threshold. The figure represents the relationship between elapsed time and: (A) Carrying EER, (B) Pushing/Pulling EER, (C) Carrying HR, and (D) Pushing/Pulling HR.

Sit-to-Stand or Functional Reach tests. However, a significant interaction was found for proprioception loss with EER (p = 0.037), indicating that for each 1 kcal/min increase in EER, proprioception decreases by 0.58°. Similarly, BMI had a significant effect (p = 0.032), with a 1 kg/m² increase in BMI associated with a 0.11° loss in proprioception. Additionally, a significant effect was found for the Sharpened Romberg test with EER (p = 0.020), suggesting that each 1 kcal/min increase in EER reduces the time a person can maintain a tandem stance by 3.85 seconds. A post-hoc power analysis indicated a power ≥ 91.3% for detecting statistical effects in balance and proprioception, considering the significant factors of EER and BMI.

Equations were developed for responses that showed significant effects. For the proprioception test, equation (2) shows the loss of knee proprioception measured by the degree difference, based on EER and BMI. A 3-D graphical representation is shown in Fig 3A.

$$\text{Loss of Proprioception } (°) = -2.16 + (0.58 \times \text{EER}) + (0.11 \times \text{BMI}) \tag{2}$$

For the Sharpened Romberg test, equation (3) shows the loss of balance measured by the time difference at the tandem position, based on EER. 2-D graphical representation is shown in Fig 3B.

**Table 3. Metabolic variables by BMI classification.** EER and HR data are reported as the average plateau value after 5-min stabilization period. Marginal means± standard deviation for BMI classifications are grouped using a Tukey pairwise comparison. A significant p-value for the ANOVA test comparing each metabolic variable with BMI classification represented with (*).

| Parameter | | Carrying | | | | | Pushing/Pulling | | | | |
|---|---|---|---|---|---|---|---|---|---|---|---|
| | | Normal (n=10) | Overweight (n=10) | Obese (n=10) | Subtotal (n=30) | P-Value | Normal (n=10) | Overweight (n=10) | Obese (n=10) | Subtotal (n=30) | P-Value |
| EER [kcal/min] | Male (n=15) | 3.54±0.25 | 4.85±0.31 | 6.49±2.77 | 4.96±1.97 | 0.003* | 5.51±1.61 | 7.18±2.06 | 8.82±1.69 | 7.17±2.17 | 0.013* |
| | Female (n=15) | 2.96±0.24 | 4.16±0.34 | 4.19±0.45 | 3.77±0.72 | | 5.55±1.02 | 5.85±0.73 | 6.56±1.29 | 5.98±1.06 | |
| | Subtotal (n=30) | 3.25±0.49 | 4.51±0.63 | 5.34±2.23 | 4.37±1.58 | | 5.53±1.27 | 6.51±1.85 | 7,67±1.62 | 6.58±1.79 | |
| | Grouping | B | AB | A | | | B | AB | A | | |
| HR [bpm] | Male (n=15) | 107.34±11.58 | 115.08±5.05 | 123.11±18.99 | 115.18±13.89 | 0.038* | 126.09±11.73 | 137.77±12.59 | 146.25±5.97 | 136.71±12.96 | 0.237 |
| | Female (n=15) | 115.50±13.69 | 128.17±17.35 | 131.91±6.81 | 125.19±14.34 | | 137.83±27.05 | 148.61±15.38 | 140.61±20.26 | 142.29±20.41 | |
| | Subtotal (n=30) | 111.43±12.70 | 111.65±13.88 | 126.51±14.22 | 120.19±14.77 | | 131.96±20.61 | 143.19±14.43 | 143.33±14.41 | 139.50±17.04 | |
| | Grouping | B | AB | A | | | A | A | A | | |

**Table 4. Summary balance and proprioception descriptive statistics per BMI classification. All data are reported as mean±standard deviation. Values represent the difference between the test measurements taken before and after execution of the MMH task.**

| MMH | Test | Normal | Overweight | Obese |
|---|---|---|---|---|
| Carrying | Proprioception (º) | 2.32±2.05 | 3.85±2.00 | 4.95±1.67 |
| | Sharpened Romberg (s) | 1.34±5.66 | −5.40±8.73 | −9.22±10.28 |
| | Functional Reach (in) | −0.23±1.40 | −0.34±1.80 | −1.19±2.08 |
| | Sit-to-Stand (#) | 0.30±2.91 | 0.60±3.56 | 0.10±1.66 |
| Pushing Pulling | Proprioception (º) | 4.14±1.73 | 3.58±2.40 | 8.29±2.38 |
| | Sharpened Romberg (s) | −8.83±15.53 | −23.57±21.72 | −17.55±13.18 |
| | Functional Reach (in) | −0.08±1.78 | −0.50±1.72 | −1.20±2.08 |
| | Sit-to-Stand (#) | −0.30±2.26 | 0.60±1.78 | 0.20±1.87 |
| Subtotal | Proprioception (º) | 3.13±2.05 | 3.72±2.16 | 6.62±2.64 |
| | Sharpened Romberg (s) | −3.75±12.52 | −14.35±17.87 | −13.38±12.27 |
| | Functional Reach (in) | −0.15±1.56 | −0.42±1.72 | −1.20±2.02 |
| | Sit-to-Stand (#) | 0.00±2.55 | 0.60±2.74 | 0.15±1.73 |

**Table 5. Results mixed effects model with random effects. The estimate (or coefficient) provides the expected effect of the predictor. T-statistics and p-value assess whether this effect is statistically significant. A significant p-value is represented with (*).**

| Fixed Effect Predictor(s) | Response | Estimate | Standard Error | t-value | p-value |
|---|---|---|---|---|---|
| Intercept | **Proprioception** (º) | −2.16 | 1.36 | −1.59 | 0.120 |
| EER | | 0.58 | 0.27 | 2.15 | 0.037* |
| BMI | | 0.11 | 0.05 | 2.21 | 0.032* |
| Sex | | 0.61 | 0.71 | 0.86 | 0.398 |
| Intercept | **Sharpened Romberg** (s) | 1.14 | 8.11 | 0.14 | 0.889 |
| EER | | −3.85 | 1.61 | −2.40 | 0.020* |
| BMI | | 0.20 | 0.33 | 0.60 | 0.551 |
| Sex | | 6.83 | 4.49 | 1.52 | 0.134 |
| Intercept | **Sit to Stand** (#) | −0.08 | 1.09 | −0.08 | 0.894 |
| EER | | −0.05 | 0.19 | −0.25 | 0.801 |
| BMI | | 0.01 | 0.04 | 0.14 | 0.893 |
| Sex | | 0.86 | 0.65 | 1.33 | 0.190 |
| Intercept | **Functional Reach** (in) | −0.31 | 0.84 | −0.37 | 0.715 |
| EER | | 0.15 | 0.15 | 1.00 | 0.320 |
| BMI | | −0.04 | 0.03 | −1.11 | 0.271 |
| Sex | | −0.05 | 0.50 | −0.11 | 0.914 |

$$\text{Time (seconds)} = 1.14 - (3.85 \times \text{EER}) \tag{3}$$

## Discussion

The 1991 NIOSH committee established a physiological criterion of 4.7 kcal/min EER to limit loads for dynamic lifting tasks (e.g., walking, load carrying, and repeated load lifting), based on the fatigue associated with performing these tasks over prolonged periods. However, these calculations were based on empirical data from the general population collected over 30 years ago [56]. In the U.S., the prevalence of obesity has nearly doubled, from 23.3% in 1988–1994 to 41.9% in 2017–2020 [57,58]. Obesity has been linked to work-related injuries, increased medical costs, and reduced work

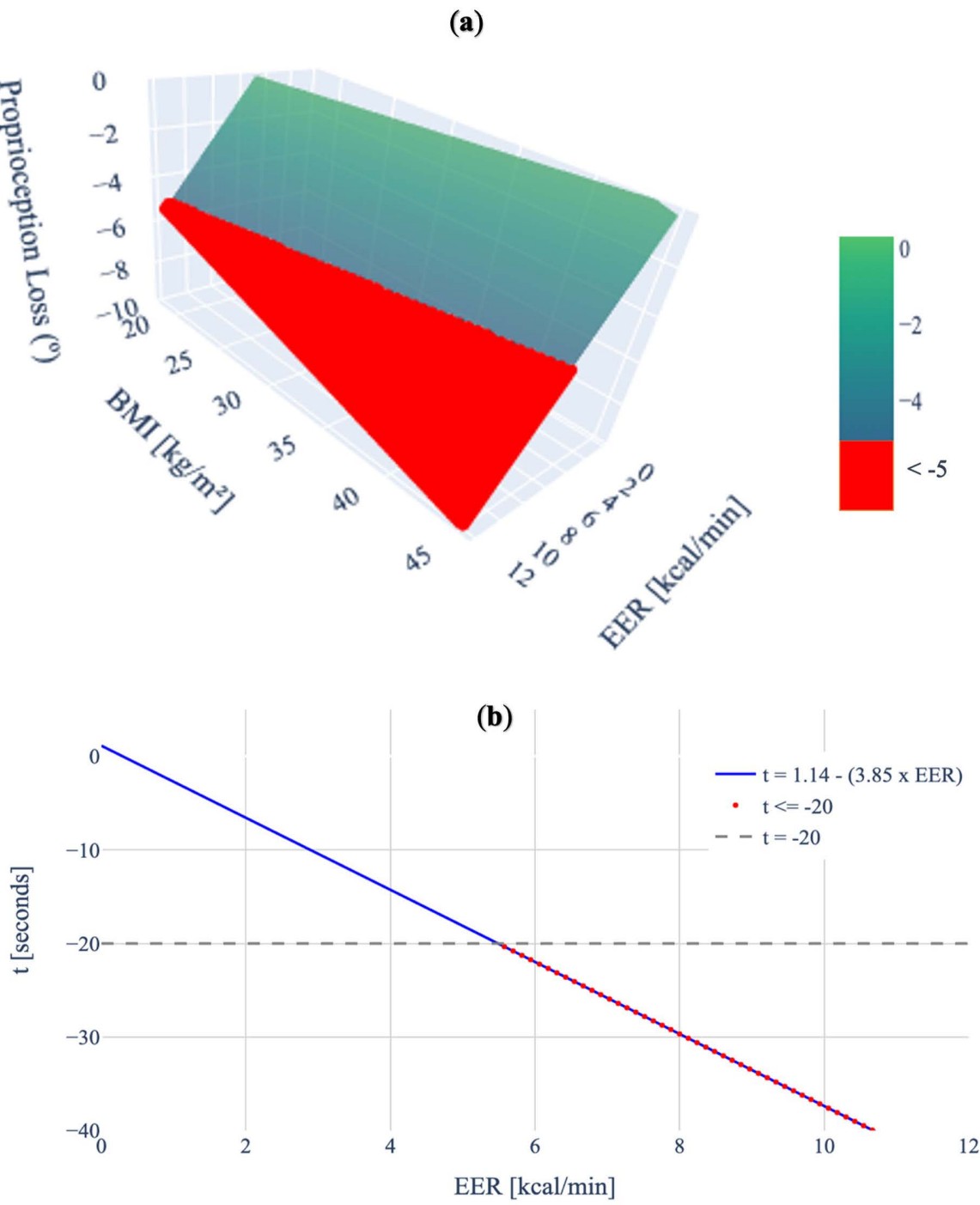

**Fig 3. Significant fixed effects for knee proprioception test and Sharpened Romberg Test.** (A) 3-D representation of the relationship between loss of proprioception, BMI, and EER. The red-colored portion of the diagram indicates regions where the loss of proprioception exceeds 5° (proprioception difference < −5°). (B) 2-D representation of the relationship between the loss of balance (measured by the time difference at the tandem position during the Sharpened Romberg test) and EER. The gray dotted line represents the threshold value of losing 20 seconds (t = −20), and the red dots indicate values under the threshold (t < −20).

productivity [59]. Despite these concerning facts, the role of obesity in physiological stress during manual material handling tasks remains unclear. To the best of the authors' knowledge, this is the first study to compare the effects of elevated BMI on physiological stress and other metabolic parameters, such as HR, during MMH tasks like carrying, pushing, and pulling, performed at MAL.

Steady-state values of EER and HR provide crucial insights into participant metabolic activity during repetitive MMH activities [60]. In the current study, the magnitudes of EER for both the carrying and pushing/pulling tasks were significantly dependent on BMI group (p = 0.003 and p = 0.013, respectively). Our previous work also reported this dependence with respect to BMI group during a repetitive manual lifting task at RWL, showing that the obese experienced higher physiological stress levels compared to non-obese groups, exceeding the safety limits established by the RNLE [24]. This is consistent with the present study's EER results for the carrying task, where only the obese group surpassed the threshold value. However, for the pushing/pulling task, all BMI classifications had EER results that exceeded the safety limit. These higher EER values for pushing/pulling tasks are perhaps due to the increased shoulder load and relatively low levels of low back loading, compared to carrying tasks, which generally have lower mechanical joint loading [61–65]. Our findings are also consistent with the lower MALs in the Liberty Mutual equations for the pushing and pulling task when compared to carrying [28]. This pattern is supported by the linear regression analysis, where EER during the carrying task showed a 68.9% positive correlation with individual BMI values, compared to a 37.6% correlation during the pushing/pulling task. These correlations build on our pervious study where a 50.8% positive correlation was observed for repetitive box lifting [24]. Overall, both MMH tasks revealed that the obese group had higher physiological stress compared to the normal-weight group, which is associated with increased risk of musculoskeletal injuries [24]. However, the pushing/pulling task resulted in higher EER values across all BMI groups compared to the carrying task, exceeding the NIOSH criteria, suggesting that pushing/pulling may be a riskier task.

Differences between the carrying and pushing/pulling tasks were also observed in the HR results. A significant correlation between BMI and HR was found for the carrying task. This supports the general trend where individuals classified as obese tend to have higher heart rates in response to exertion [66–68]. Similarly, in the linear regression analyses, a significant correlation between individual BMI values and HR was observed for carrying. This aligns with our previous study, which found a 21.8% significant positive correlation between BMI and HR [24] during symmetrical lifting tasks. Notably, 80% of obese individuals exceeded the NIOSH threshold. In contrast, during the pushing/pulling task, most participants, regardless of BMI, exceeded the NIOSH threshold. This suggests that the relationship between BMI and HR may be one where obese individuals are more likely to exceed the NIOSH threshold and experience elevated HR during MMH tasks. However, the HRs recorded during the pushing/pulling task also indicate that when participants exceed the NIOSH threshold, the differences in elevated HR between BMI classifications become less apparent.

Sex also influenced physiological stress for both MMH tasks, with males exhibiting higher EER than females by 31.6% for carrying and 19.9% for pushing/pulling. While Klausen et al. found no difference between male and female EER during general exercise [69], a recent systematic review reported that physiological differences between males and females during carrying tasks are mainly due to anthropometric differences rather than biological factors [70]. This finding aligns with our study, as female BMI was lower in both the obese and normal BMI groups, likely affecting the EER results. Therefore, increased conditioning and lean body mass may be more important factors than sex for elevated values of HR during MMH tasks [71,72].

In the present study, the EER of obese participants exceeded the NIOSH physiological for MMH over prolonged periods. As a result, physiological stress levels in obese workers are reaching concerning levels due to fatigue. Fatigue-related impaired balance is a key factor associated with an increased risk of falls [13,14]. Additionally, obesity itself is known to affect postural stability and balance [73,74]. However, no previous study has quantified the extent to which elevated BMI and EER contribute to the loss of balance and proprioception. Established balance and proprioception tests have been major indicators of risk for falls [37,38,40,41,75]. Out of the four tests performed by participants before and

after the MMH tasks (knee proprioception, Sharpened Romberg, Functional Reach, and Sit-to-Stand), significant interactions were observed only in the knee proprioception and Sharpened Romberg test.

For loss of knee proprioception, both BMI and EER had a significant positive effect, as shown in Fig 3A. Previous studies have shown that a deviation of 3–5º in knee proprioception is a critical threshold that correlates with an increased risk of injury and falls [36,76,77]. Based on the developed formula, when the 4.7 kcal/min limit is not exceeded, only a BMI greater than 40.3 surpasses the 5º proprioception threshold for increased risk of injury. This indicates that the NIOSH physiological safety criteria are effective for the general population when assuming that their EER is under the NIOSH threshold. However, as previously shown, obese individuals can surpass this limit. According to the developed formula, participants with a normal BMI would be at risk of injury or falling at an EER of 7.6 kcal/min, those who are overweight at 6.7 kcal/min, and those who are obese at 5.7 kcal/min. In the present study, 65% of obese participants exceeded 5.7 kcal/min, putting them at risk of injury or falls, in contrast to only 25% of the participants for both overweight and normal BMI groups.

For the Sharpened Romberg test, only EER had a positive significant effect in the mixed-effect model, as shown in Fig 3B. Agrawal et al. reported that the risk of falling increases more than threefold if a participant's balance time is less than 20 seconds [78]. Based on the developed formula and using the NIOSH safety criteria, assuming the 4.7 kcal/min threshold is not exceeded, this 20-second threshold is generally exceeded, further indicating that the NIOSH threshold is suitable for the general population. However, if balance time falls below this 20-second threshold, participants with an EER of 5.49 kcal/min or higher were found to have a threefold increased risk of falling. Remarkably, this value is similar to the one found in loss of knee proprioception (5.7 kcal/min). In the present study, 65% of obese participants exceeded 5.49 kcal/min, compared to only 40% of overweight participants and 25% of those with normal BMI.

Some limitations should be acknowledged. A total of 30 participants (15 male and 15 female) were evaluated in the present study. While a larger sample size would have been preferable, results from the post-hoc power analysis indicated that the sample was sufficient to detect significant differences in EER and HR (power ≥ 95%) across BMI groups and sex. Additionally, the sample was adequate for detecting effects in balance/proprioception values (power ≥ 91.3%), considering EER and BMI as the main effects. In addition, the participants' job roles did not consistently involve MMH tasks. Future research should specifically examine the effects of physiological stress, balance, and proprioception on participants that perform MMH tasks as part of their occupation to draw more definitive conclusions. Furthermore, BMI does not account for body fat percentage or its distribution. Although the researchers selected only obese participants with minimal to no reported exercise history, body fat percentage, which is a more precise indicator of physiological stress than BMI, was not directly measured. However, as stated in the Results section, waist circumference showed an almost perfect correlation with BMI, suggesting that the participants' BMI in this study is closely related to their body fat percentage [55] and can be used as a substitute. Finally, this study explored the impact of obesity on physiological stress and loss of balance and proprioception, using 77–79% of the female percentile as defined by the psychophysical NIOSH criterion [26]. Future studies could quantify EER and fall risk in percentiles above 80% for females to determine whether BMI's significant role persists when workers handle lighter loads during repetitive manual tasks.

In summary, the present study showed that obese individuals surpassed the physiological NIOSH criterion for both carrying and pushing/pulling tasks and the same metabolic stress is higher during pushing and pulling than carrying. Our study also showed that obese individuals have impaired proprioception and balance while performing repetitive MMH tasks. Additionally, the present study introduces a way to quantify the loss of proprioception and balance through an empirical relationship of BMI and EER.

## Supporting information

**S1 Fig. Relationships between metabolic variables and BMI.** Each circle represents a participant, with colors indicating BMI classification: blue for obese, green for overweight, and purple for normal. The solid red line red represents the 4.7 kcal/min NIOSH threshold. r² and p-values are provided for each graph. The figure represents the relationship

between a subject's BMI and the following variables: (A) Carrying EER; (B) Carrying HR; (C) Pushing/pulling EER; and (D) Waist circumference.
(TIF)

**S1 Table. Source data.** Individual data points for metabolic data (EER and HR) and balance and proprioception tests. Participants are categorized by BMI classification.
(XLSX)

## Author contributions

**Conceptualization:** Sergio A. Lemus, Jaron Mohammed, Eduard Tiozzo, Francesco Travascio.

**Data curation:** Sergio A. Lemus, Jaron Mohammed, Cheng-Bang Chen, Francesco Travascio.

**Formal analysis:** Sergio A. Lemus, Jaron Mohammed, Cheng-Bang Chen, Francesco Travascio.

**Funding acquisition:** Francesco Travascio.

**Investigation:** Sergio A. Lemus, Jaron Mohammed, Thomas M. Best, Eduard Tiozzo, Francesco Travascio.

**Methodology:** Sergio A. Lemus, Jaron Mohammed, Thomas M. Best, Eduard Tiozzo, Francesco Travascio.

**Writing – original draft:** Sergio A. Lemus, Jaron Mohammed, Cheng-Bang Chen, Thomas M. Best, Eduard Tiozzo, Francesco Travascio.

**Writing – review & editing:** Sergio A. Lemus, Jaron Mohammed, Cheng-Bang Chen, Thomas M. Best, Eduard Tiozzo, Francesco Travascio.

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
