## [Decision Letter · Decision Letter 0]

18 Feb 2025

PONE-D-25-03867THE ROLE OF OBESITY IN PHYSIOLOGICAL STRESS, BALANCE, AND PROPRIOCEPTION DURING REPETITIVE MANUAL MATERIAL HANDLING TASKSPLOS ONE

Dear Dr. Travascio,

Thank you for submitting your manuscript to PLOS ONE. After careful consideration, we feel that it has merit but does not fully meet PLOS ONE’s publication criteria as it currently stands. Therefore, we invite you to submit a revised version of the manuscript that addresses the points raised during the review process.

We look forward to receiving your revised manuscript.

Kind regards,

Emiliano Cè, Ph.D.

Academic Editor

PLOS ONE

Journal Requirements:

“This work was supported by the University of Miami under the Laboratory for Integrative Knowledge (U-LINK) Predoctoral Fellowship in Interdisciplinary Research (Award Number U-LINK 23-4095) and the 2024 Summer Academic Enhancement Research Fellowship”

Additional Editor Comments:

Dear Authors,

one expert in the field reviewed your manuscript reporting several minor issues you should consider during the revision process. 

Reviewers' comments:

Reviewer's Responses to Questions

**Comments to the Author**

1. Is the manuscript technically sound, and do the data support the conclusions?

Reviewer #1: Partly

2. Has the statistical analysis been performed appropriately and rigorously? 

Reviewer #1: Yes

3. Have the authors made all data underlying the findings in their manuscript fully available?

Reviewer #1: Yes

4. Is the manuscript presented in an intelligible fashion and written in standard English?

Reviewer #1: Yes

5. Review Comments to the Author

Reviewer #1: The manuscript presents a technically sound study on the effects of obesity on physiological stress, balance, and proprioception during manual material handling tasks. The methodology is rigorous, and the statistical analysis is appropriate, though the small sample size (n=30) and reliance on BMI as an obesity measure may limit generalizability. Expanding the sample size, incorporating direct body composition assessments, and recruiting occupation-specific participants would enhance validity. The study is clearly written, with well-structured arguments and practical implications for workplace ergonomics. Minor revisions are recommended to improve external applicability and strengthen conclusions.

6. PLOS authors have the option to publish the peer review history of their article (what does this mean? ). If published, this will include your full peer review and any attached files.

**Do you want your identity to be public for this peer review?** For information about this choice, including consent withdrawal, please see our Privacy Policy .

Reviewer #1: **Yes: ** Anees Alyafei

---

## [Author Response · Author response to Decision Letter 0]

19 Feb 2025

AResponses to Reviewers’ Comments

Reviewer: Dr. Anees Alyafei

Review Comments to the Author

Reviewer #1: The manuscript presents a technically sound study on the effects of obesity on physiological stress, balance, and proprioception during manual material handling tasks. The methodology is rigorous, and the statistical analysis is appropriate, though the small sample size (n=30) and reliance on BMI as an obesity measure may limit generalizability. Expanding the sample size, incorporating direct body composition assessments, and recruiting occupation-specific participants would enhance validity. The study is clearly written, with well-structured arguments and practical implications for workplace ergonomics. Minor revisions are recommended to improve external applicability and strengthen conclusions.

Response: We sincerely appreciate the reviewer's insightful comments and valuable suggestions. In response to his recommendations, we have revised the manuscript accordingly. Please find our detailed responses below, with revisions highlighted in yellow.

In determining the total number of participants for this study (30 participants), a power analysis was conducted to ensure statistical robustness and meaningful results. This consideration has been incorporated into the Discussion section. Changes have been highlighted in lines 344-349: “A total of 30 participants (15 male and 15 female) were evaluated in the present study. While a larger sample size would have been preferable, results from the post-hoc power analysis indicated that the sample was sufficient to detect significant differences in EER and HR (power ≥ 95%) across BMI groups and sex. Additionally, the sample was adequate for detecting effects in balance/proprioception values (power ≥ 91.3%), considering EER and BMI as the main effects.”

Regarding the incorporation of direct body composition assessment, the limitations of BMI and the use of waist circumference as a substitute were addressed in the Discussion section. These changes have been added and highlighted in lines 352-358: “Furthermore, BMI does not account for body fat percentage or its distribution. Although the researchers selected only obese participants with minimal to no reported exercise history, body fat percentage, which is a more precise indicator of physiological stress than BMI, was not directly measured. However, as stated in the Results section, waist circumference showed an almost perfect correlation with BMI, suggesting that the participants' BMI in this study is closely related to their body fat percentage (Bozeman et al. 2012) and can be used as a substitute.”

Lastly, the limitation of recruiting occupation-specific participant was acknowledged in the Discussion section. Changes have been added and highlighted in lines 349-352: “In addition, the participants' job roles did not consistently involve MMH tasks. Future research should specifically examine the effects of physiological stress, balance, and proprioception on participants that perform MMH tasks as part of their occupation to draw more definitive conclusions.”

---

## [Decision Letter · Decision Letter 1]

28 Mar 2025

PONE-D-25-03867R1THE ROLE OF OBESITY IN PHYSIOLOGICAL STRESS, BALANCE, AND PROPRIOCEPTION DURING REPETITIVE MANUAL MATERIAL HANDLING TASKSPLOS ONE

Dear Dr. Travascio,

Thank you for submitting your manuscript to PLOS ONE. After careful consideration, we feel that it has merit but does not fully meet PLOS ONE’s publication criteria as it currently stands. Therefore, we invite you to submit a revised version of the manuscript that addresses the points raised during the review process.

**ACADEMIC EDITOR: **Dear Authors, your R1-manuscript version was re-assigned to two experts in the field that still reported some minor points you should consider during the revision process.============================== Please submit your revised manuscript by May 12 2025 11:59PM. If you will need more time than this to complete your revisions, please reply to this message or contact the journal office at plosone@plos.org . Please include the following items when submitting your revised manuscript:

We look forward to receiving your revised manuscript.

Kind regards,

Emiliano Cè, Ph.D.

Academic Editor

PLOS ONE

Journal Requirements:

Reviewers' comments:

Reviewer's Responses to Questions

**Comments to the Author**

1. If the authors have adequately addressed your comments raised in a previous round of review and you feel that this manuscript is now acceptable for publication, you may indicate that here to bypass the “Comments to the Author” section, enter your conflict of interest statement in the “Confidential to Editor” section, and submit your "Accept" recommendation.

Reviewer #1: All comments have been addressed

Reviewer #2: All comments have been addressed

2. Is the manuscript technically sound, and do the data support the conclusions?

Reviewer #1: Partly

Reviewer #2: Yes

3. Has the statistical analysis been performed appropriately and rigorously? 

Reviewer #1: Yes

Reviewer #2: Yes

4. Have the authors made all data underlying the findings in their manuscript fully available?

Reviewer #1: Yes

Reviewer #2: Yes

5. Is the manuscript presented in an intelligible fashion and written in standard English?

Reviewer #1: Yes

Reviewer #2: Yes

6. Review Comments to the Author

Reviewer #1: The study employs a rigorous experimental design, including controlled testing conditions for balance, proprioception, and physiological stress.

Appropriate statistical methods (two-way ANOVA, mixed-effects models) were used to compare groups.

Energy Expenditure Rate (EER) and Heart Rate (HR) as physiological markers are well-chosen and valid indicators of fatigue and stress.

The study follows ethical research guidelines, with Institutional Review Board (IRB) approval and informed consent obtained.

Small Sample Size (n=30): Although a post-hoc power analysis justifies the sample size, a larger sample would improve generalizability.

BMI is used as the primary obesity classification metric. While the authors include waist circumference, more precise.

Long and Complex Sentence Structures:

Original:

"Manual laborers are prone to fatigue-related incidents, increasing the risk of balance disturbances and falls. Our previous work showed that obese individuals can reach critical fatigue levels during repetitive lifting, as defined by National Institute for Occupational Safety and Health (NIOSH). Therefore, assessing fatigue levels during manual handling activities beyond lifting is crucial to determine if this body mass index (BMI)-based pattern persists."

Issue: The sentence is long and contains multiple ideas that could be broken down for clarity.

Suggested Revision:

"Manual laborers often experience fatigue-related incidents, which increase their risk of balance disturbances and falls. Previous research indicates that obese individuals may reach critical fatigue levels during repetitive lifting. This study examines whether this BMI-based fatigue pattern also applies to other manual handling activities."

Original:

"Additionally, evaluating balance loss when physiological stress becomes critical will help quantify the increased fall risk for obese individuals."

Issue: The phrase "evaluating balance loss when physiological stress becomes critical" is somewhat ambiguous.

Suggested Revision:

"Assessing balance impairment under high physiological stress conditions will help quantify the increased fall risk in obese individuals."

⚠ Minor Typographical and Grammatical Errors:

Original:

"Results indicated a positive correlation between BMI and EER for both carrying (p = 0.003) and pushing/pulling (p = 0.013). In the mixed-effects model, BMI (p = 0.032) and EER (p = 0.037) were positively correlated with knee proprioception loss, whereas EER was negatively correlated with balance (p = 0.020). These findings confirm that obese individuals face a critical fatigues levels and impaired proprioception and balance during repetitive handling tasks."

Issue: "face a critical fatigues levels" contains a grammatical error ("fatigues" should be "fatigue").

Suggested Revision:

"These findings confirm that obese individuals face critical fatigue levels, as well as impaired proprioception and balance, during repetitive handling tasks."

Original:

"Carrying and pushing/pulling tasks were performed by 30 participants, with weights calculated using the Liberty Mutual Equations to meet NIOSH criteria."

Issue: "Carrying and pushing/pulling tasks were performed by 30 participants" is slightly awkward in construction.

Suggested Revision:

"Thirty participants performed carrying and pushing/pulling tasks, with weights determined using the Liberty Mutual Equations to align with NIOSH criteria."

Reviewer #2: (No Response)

7. PLOS authors have the option to publish the peer review history of their article (what does this mean? ). If published, this will include your full peer review and any attached files.

**Do you want your identity to be public for this peer review?** For information about this choice, including consent withdrawal, please see our Privacy Policy .

Reviewer #1: **Yes: ** Anees Alyafei

Reviewer #2: No

---

## [Author Response · Author response to Decision Letter 1]

30 Mar 2025

Responses to Reviewers’ Comments

Reviewers: Anees Alyafei, Reviewer #2

Review Comments to the Author

Reviewer #1:

The study employs a rigorous experimental design, including controlled testing conditions for balance, proprioception, and physiological stress. Appropriate statistical methods (two-way ANOVA, mixed-effects models) were used to compare groups. Energy Expenditure Rate (EER) and Heart Rate (HR) as physiological markers are well-chosen and valid indicators of fatigue and stress. The study follows ethical research guidelines, with Institutional Review Board (IRB) approval and informed consent obtained.

Small Sample Size (n=30):

Although a post-hoc power analysis justifies the sample size, a larger sample would improve generalizability. BMI is used as the primary obesity classification metric. While the authors include waist circumference, more precise.

Response: We sincerely appreciate the reviewer's feedback and valuable suggestions. In response to his recommendations, we have revised the manuscript accordingly. Please find our detailed responses below, with revisions highlighted in yellow.

Regarding the sample size of 30 participants, we acknowledged this as a limitation in the Discussion section, mentioning the inclusion of a post-hoc power analysis and that a larger sample size would be ideal. This was included and highlighted in lines 342-347: “A total of 30 participants (15 male and 15 female) were evaluated in the present study. While a larger sample size would have been preferable, results from the post-hoc power analysis indicated that the sample was sufficient to detect significant differences in EER and HR (power ≥ 95%) across BMI groups and sex. Additionally, the sample was adequate for detecting effects in balance/proprioception values (power ≥ 91.3%), considering EER and BMI as the main effects.”

Long and Complex Sentence Structures:

Original: "Manual laborers are prone to fatigue-related incidents, increasing the risk of balance disturbances and falls. Our previous work showed that obese individuals can reach critical fatigue levels during repetitive lifting, as defined by National Institute for Occupational Safety and Health (NIOSH). Therefore, assessing fatigue levels during manual handling activities beyond lifting is crucial to determine if this body mass index (BMI)-based pattern persists."

Issue: The sentence is long and contains multiple ideas that could be broken down for clarity.

Suggested Revision: "Manual laborers often experience fatigue-related incidents, which increase their risk of balance disturbances and falls. Previous research indicates that obese individuals may reach critical fatigue levels during repetitive lifting. This study examines whether this BMI-based fatigue pattern also applies to other manual handling activities."

Response: We have addressed it as suggested and incorporated the following revisions in lines 002–005: “Manual laborers often experience fatigue-related incidents, which increase their risk of balance disturbances and falls. Previous research indicates that obese individuals may reach critical fatigue levels during repetitive lifting. This study examines whether this BMI-based fatigue pattern also applies to other manual handling activities.”

Original: "Additionally, evaluating balance loss when physiological stress becomes critical will help quantify the increased fall risk for obese individuals."

Issue: The phrase "evaluating balance loss when physiological stress becomes critical" is somewhat ambiguous.

Suggested Revision: "Assessing balance impairment under high physiological stress conditions will help quantify the increased fall risk in obese individuals."

Response: Suggested revisions have been added and highlighted in lines 005-007: “Therefore, assessing balance impairment under high physiological stress conditions will help quantify the increased fall risk in obese individuals.”

Minor Typographical and Grammatical Errors:

Original: "Results indicated a positive correlation between BMI and EER for both carrying (p = 0.003) and pushing/pulling (p = 0.013). In the mixed-effects model, BMI (p = 0.032) and EER (p = 0.037) were positively correlated with knee proprioception loss, whereas EER was negatively correlated with balance (p = 0.020). These findings confirm that obese individuals face a critical fatigues levels and impaired proprioception and balance during repetitive handling tasks."

Issue: "face a critical fatigues levels" contains a grammatical error ("fatigues" should be "fatigue"). Suggested Revision: "These findings confirm that obese individuals face critical fatigue levels, as well as impaired proprioception and balance, during repetitive handling tasks."

Response: We have corrected this sentence and highlighted the changes in lines 014–016: “These findings confirm that obese individuals face critical fatigue levels, as well as impaired proprioception and balance, during repetitive handling tasks.”

Original: "Carrying and pushing/pulling tasks were performed by 30 participants, with weights calculated using the Liberty Mutual Equations to meet NIOSH criteria." Issue: "Carrying and pushing/pulling tasks were performed by 30 participants" is slightly awkward in construction.

Suggested Revision: "Thirty participants performed carrying and pushing/pulling tasks, with weights determined using the Liberty Mutual Equations to align with NIOSH criteria.

Response: Suggested revision was added and highlighted in lines 007–008: “Thirty participants performed carrying and pushing/pulling tasks, with weights determined using the Liberty Mutual Equations to align with NIOSH criteria.”

Reviewer #2: (No Response)

---

## [Editor Report · Decision Letter 2]

5 May 2025

THE ROLE OF OBESITY IN PHYSIOLOGICAL STRESS, BALANCE, AND PROPRIOCEPTION DURING REPETITIVE MANUAL MATERIAL HANDLING TASKS

PONE-D-25-03867R2

Dear Dr. Travascio,

We’re pleased to inform you that your manuscript has been judged scientifically suitable for publication and will be formally accepted for publication once it meets all outstanding technical requirements.

Kind regards,

Emiliano Cè, Ph.D.

Academic Editor

PLOS ONE
---

## [Editor Report · Acceptance letter]

PONE-D-25-03867R2

PLOS ONE

Dear Dr. Travascio,

I'm pleased to inform you that your manuscript has been deemed suitable for publication in PLOS ONE. Congratulations! Your manuscript is now being handed over to our production team.

Kind regards,

on behalf of

Prof. Emiliano Cè

Academic Editor

PLOS ONE